# Proteomic Analysis of Fresh and Liquid-Stored Boar Spermatozoa

**DOI:** 10.3390/ani10040553

**Published:** 2020-03-26

**Authors:** Blanka Premrov Bajuk, Petra Zrimšek, Maja Zakošek Pipan, Bruno Tilocca, Alessio Soggiu, Luigi Bonizzi, Paola Roncada

**Affiliations:** 1Institute of Preclinical Sciences, Veterinary Faculty, University of Ljubljana, Gerbičeva 60, 1000 Ljubljana, Slovenia; petra.zrimsek@vf.uni-lj.si; 2Clinic for Reproduction and Large Animals, Veterinary Faculty, University of Ljubljana, Gerbičeva 60, 1000 Ljubljana, Slovenia; maja.zakosekpipan@vf.uni-lj.si; 3Department of Health Sciences, University ‘Magna Græcia´of Catanzaro, Viale Europa, 88100 Catanzaro, Italy; tilocca@unicz.it (B.T.); roncada@unicz.it (P.R.); 4Department of Veterinary Medicine, Università degli Studi di Milano, Via dell‘Università 6, 26900 Lodi, Italy; alessio.soggiu@unimi.it; 5Department of Biomedical, Surgical and Dental Sciences, Università degli Studi di Milano, Via della Commenda 10, 20122 Milano, Italy; Luigi.Bonizzi@unimi.it

**Keywords:** liquid stored semen, boar spermatozoa, proteomics

## Abstract

**Simple Summary:**

Male gametes can be stored for a long period of time for the purpose of preserving genetic material. Cryopreservation and liquid preservation are two main storage procedures commonly used for boar semen. There is evidence in the literature suggesting that cryopreservation changes the profile of proteins that are linked to the motility and function of spermatozoa. It was postulated that they can affect motility, capacitation, oocyte binding ability, and the acrosome reaction of spermatozoa. On the other hand, little is known about changes in protein levels in sperm that occur during liquid storage. Therefore, the objective of this study was to investigate whether liquid storage also causes an alteration in the proteomic profile of stored spermatozoa. A comparative proteomic approach was used to analyze protein samples from fresh spermatozoa and spermatozoa stored for three days at 15–17 °C. Results obtained show that liquid preservation causes quantitative changes in the boar sperm proteome with the over-expression of three out of four proteins in the liquid-stored sperm. Our findings can help elucidate the events involved in liquid preservation.

**Abstract:**

In this study comparative proteomics was used to define changes in the expression of the spermatozoa proteins during liquid storage. Semen from eight boars was analyzed on the day of collection and after liquid preservation at 15–17 °C for three days. Sperm parameters (concentration, motility, morphology, vitality) and percentage of non-capacitated and acrosomal-reacted spermatozoa were determined. Sperm proteins were extracted and separated by two-dimensional sodium dodecyl sulphate-polyacrylamide gel electrophoresis (SDS-PAGE) and proteomic profiles were computationally compared to highlight differentially expressed protein spots that were, in turn, identified by mass spectrometry. The intensities of four spots were significantly different between fresh and liquid stored sperm. Namely: ATP citrate lyase, chaperonin containing T-complex polypeptide 1 (TCP1) subunit ε and probable phospholipid-transporting ATP-ase were over-expressed in liquid stored sperm, whereas cytosolic non-specific dipeptidase was over-expressed in fresh sperm. These differentially expressed proteins could be used as plausible biomarkers for the evaluation of boar semen quality and spermatozoa survival after liquid storage and could help to address problems associated with sperm preservation.

## 1. Introduction

Artificial insemination (AI) with extended semen offers many benefits to the swine industry through improving biosecurity and access to high-quality genetic material [1]. The modern boar industry worldwide is based on the use of AI of sows with extended seminal doses stored at 15–20 °C for 1–5 days, but usually used within one day. When Beltsville thawing solution (BTS), a so-called short-term extender is used, AI usually takes place within 72 h after semen collection because longer storage is commonly associated with a decreased litter size [2]. Semen preservation has many negative effects on spermatozoa which are related to dilution, change of microenvironment, ageing and chilling [3]. Low temperatures and extender composition help decrease sperm metabolic activity, but at the same time maintain their function and fertilizing ability. Several changes occur during liquid storage of boar semen, including a decrease in sperm motility, viability, and plasma membrane stability as well as an increase of oxidative stress (OS), lipid peroxidation, and apoptotic-like events [4]. Conventional semen evaluation for AI generally includes a measure of seminal volume, sperm concentration, viability, motility and morphology [5]. Some of these parameters were correlated with farrowing rate or litter size [6], but they may not accurately detect altered/non-functional spermatozoa within boar ejaculates kept in refrigeration that may result in lower reproductive performance [7]. Moreover, boar sperm longevity and survivability varies among individuals [8]. Despite the optimal quality assessed before cooling, ejaculates may respond differently to liquid preservation [9]. Predicting the fertility outcome of cooled or frozen spermatozoa is one of the most relevant aspects in the field of porcine reproduction [5]. A recently published review summarizes the current knowledge about boar seminal plasma and the sperm proteome, focusing mainly on its relevance to sperm cryopreservation procedures and their outcomes in terms of sperm function and fertility [10]. With the use of comprehensive Multidimensional Protein Identification Technology (MudPIT) shotgun proteomic analysis a better understanding of mature boar spermatozoa was provided. It also made it possible to identify vast number of proteins that were classified by their molecular functions, involvement in biological processes and participation in relevant metabolic pathways associated with spermatozoa physiology, fertility potential and protection [11]. Nevertheless, investigations on the potential alteration of the liquid stored boar sperm are missing. In Slovenia, the majority of semen AI doses are distributed as liquid-stored and used within three days. Therefore, it was the objective of the present study to investigate whether liquid storage of boar sperm for three days reflects the proteomics alteration observed in the cryopreserved spermatozoa. For this purpose, two-dimensional gel electrophoresis (2-DE) followed by the mass fingerprinting (matrix-assisted laser desorption/ionization time-of-flight mass spectrometry (MALDI-TOF/MS)) approach were employed to evaluate the protein profile of stored sperm samples relative to that of fresh sperm samples from the same ejaculates. The proteins identified might allow a better comprehension of the events involved in liquid sperm preservation and may also help to discover new fertility and sperm performance biomarkers.

## 2. Materials and Methods 

### 2.1. Semen Samples and Analysis

According to Slovenian legislation (National Assembly of the Republic of Slovenia, 2013) which is covered by Directive 2010/63/EU (The European Parliament & the Council, 2010) the ethical approval for the experiments was not needed because this study employed only non-invasive procedures. 

Ten semen samples from eight 12- to 24-month-old boars of various breeds (three Slovenian landrace, two Slovenian large white, two Pietrain, and one Hibride line) and of proven fertility that are routinely used as semen donors for artificial insemination at the local AI centre were collected using the glove-hand technique. If two sperm samples were taken from the same boar, there was at least one week time interval between the collections. Gel, dust and bristles were filtered out by a semen-collecting flask. Filtered semen of each ejaculate was than extended with Betsville Thawing Solution (BTS, Truadeco, Netherlands) at a ratio of 1:2. Computer assisted semen analysis (CASA) with a Makler counting chamber was used to determine sperm concentration and motility characteristics. Before analysis, semen samples were incubated in a water bath (37 °C) for 8 min. After that, for each sample, 5 µL of diluted semen was mounted on a warm Makler counting chamber (Sefi Medical Instruments, Israel). Three randomly selected microscopic fields were scanned three times each, obtaining nine scans for every semen sample. The mean of three scans for each microscopic field was used for the statistical analysis. The software settings for CASA (Hamilton Thorne IVOS 10.2, Hamilton Thorne Research, Beverly, MA, USA) used in this study were as follows: frames per second: 60 Hz; minimum cell size: 10 pixels; cell intensity: 125; straightness (STR) threshold: 80%; medium average path velocity (VAP) cut-off: 45 µm/s; low straight line velocity (VSL) cut-off: 15 µm/s; low VAP cut-off: 25 µm/s; slow cells motile: no; static head intensity: 0.65–4.90; static intensity gate: 0.50–2.50; magnification: 1.89; temperature of analysis: 37 °C. Sperm viability was assessed using Hoechst staining as previously described [12]. Briefly, 5 µL of bis-benzimidazole solution (Hoechst 33258, Merck, Darmstadt, Germany) was added to 100 µL of semen samples and incubated in the dark at 39 °C and 5% CO_2_ for 15 min. Five µL of stained semen samples were then placed on the Makler chamber and analysed with the CASA system. At least 600 spermatozoa per sample were analyzed in duplicates and the mean percentage of live and dead spermatozoa was determined. Following the fixation in Giemsa stain the morphology of 200 sperms in diluted semen samples was assessed in duplicates [13]. Spermatozoa were assessed using a light microscope (Olympus BH-2, Hamburg, Germany), objective (Olympus, 100x, S Plan 100 PL 160/0.17) with 1000× magnification. In order to identify the percentage of capacitated-like and acrosome reacted spermatozoa chlortetracycline staining was used [14]. One hundred spermatozoa in each preparation were assessed under the Olympus BX40 microscope equipped with a 400–440 nm excitation filter, a 475 nm emission filter and a 455 nm dichromatic mirror. Capacitation status was classified as un-capacitated (UC), capacitated-like (C) or acrosome reacted (AR). 

Ten semen samples were then stored for three days at 15–17 °C and, during storage constant, gentle agitation of semen samples was performed. After three days of liquid storage, semen characteristics were evaluated again using the same procedures as stated above. Statistical tests were performed using SigmaStat 3.5 (SYSTAT Software Inc., San Jose, USA) software. Data are presented as mean ± standard deviation (SD). Normal distribution of the data was tested by the Shapiro–Wilk test. Statistical comparison of the results obtained on days 0 and 3 (after liquid storage) for each semen parameter was performed with a *t*-test in the case of normal distribution of the data or with the Wilcoxon signed-rank test if the distribution of the data was not normal. *P* < 0.05 was considered as significant. No comparison of reproductive outcome was performed between insemination with fresh and liquid stored semen.

### 2.2. The Extraction of Spermatozoa Proteins

Protein extraction was performed on in total 20 semen samples (10 samples on the day of collection and 10 samples after storage). Extender debris and dead spermatozoa were removed from 1 mL of semen with the discontinuous Percoll density gradient (2 mL of 90% and 2 mL of 45% Percoll). The samples were centrifuged for 30 min at 700× *g* at room temperature. The spermatozoa pellets were washed three times with 4 mL of Beltsville Thawing Solution (BTS) and centrifuged for 15 min at 1000× *g* at room temperature. The pellets were transferred into a 1.5 mL Eppendorf tube, washed twice with 500 μL of BTS and centrifuged for 15 min at 1000× *g* at room temperature. Approximately 50 µL of spermatozoa was obtained and resuspended in 480 µL of extraction buffer (7 M urea, 2 M thiourea, 4% CHAPS, 1% dithiothreitol (DTT), 1/100 (*v*/*v*) protease inhibitor cocktail (Sigma, Steinheim, Germany), 1/100 (*v*/*v*) phosphatase inhibitor cocktail 2 (Sigma) and 1/100 (*v*/*v*) phosphatase inhibitor cocktail 3 (Sigma)) to extract proteins. The solution was sonicated three times for 20 seconds at full power in ice bath. Samples were then incubated at constant agitation in an Eppendorf tube with a micromagnet for 1.5 h at room temperature. After centrifugation for 30 min at 1400× *g* at 20 °C the protein concentration was measured and supernatant aliquoted and frozen at −80 °C until further analysis. 

#### Protein Quantification

The DC Protein Assay (Bio-Rad, GmbH, Germany) was used to determine the protein concentration of each sample in duplicates. The absorbance was measured at 750 nm. Bovine serum albumin (BSA, Sigma) with a concentration range between 0.2 and 1.4 mg/mL was used to plot a standard curve. A linear regression equation was applied to determine the total protein concentration in the boar spermatozoa extracts.

### 2.3. Gel Electrophoresis

The protein profile of spermatozoa extracts was determined using two-dimensional gel electrophoresis. Four pools of protein extracts from 10 different fresh and four pools of protein extracts from 10 different stored sperm samples were prepared. Each pool included 5 protein extracts, randomly prepared from 10 different samples. In total, 8 pools of extracted proteins were prepared for proteomic analysis. To run the first dimension (isoelectric focusing), 7-cm long non-linear pH range 3–10 Immobiline DryStrip (GE Healthcare) were rehydrated in 125 µL of protein solution (containing a total of approximately 95 µg of protein) for 4 h at room temperature. Isoelectric focusing (IEF) was performed using a Protean IEF Cell (Bio-Rad) according to the following settings: 30 min at 100 V, 30 min at 500 V, 30 min at 1000 V, 30 min at 2500 V and at 4000 V until 40,000 V/h were reached in total. After focusing, the strips were incubated in equilibration buffer (50 mM Tris-HCl, 6 M urea, 30% (*v*/*v*) glycerol, 2% (*w*/*v*) sodium dodecyl sulphate (SDS), 0.002% (*w*/*v*) bromophenol blue, 1% (*w*/*v*) DTT) with gentle shaking for 15 min followed by 15 min incubation in 2.5% iodoacetamide in the same buffer. After equilibration, strips were placed on the top of sodium dodecyl sulphate-polyacrylamide gel electrophoresis (SDS-PAGE) gels (10%) sealed with agarose sealing solution (0.5% agarose, 0.002% (*w*/*v*) bromophenol blue). The electrophoretic run was performed until the dye front reached the gel bottom. Gels were incubated in a colloidal Coomassie G250 solution (Sigma) and de-stained in MiliQ water.

#### Image Scan and Data Analysis

Stained gels were scanned with a calibrated densitometer ImageScanner III (GE Healthcare) and images analyzed using Progenesis SameSpot v. 4.5 (Non-linear Dynamics, UK). Images processing was performed as described previously [15]. A paired *t*-test (SAS JMP pro 14, SAS Institute srl, Milan, Italy) was used to compare the normalized protein spot volumesof fresh sperm and sperm after three days of storage. Differentially expressed spots with *P* < 0.05 were selected for the identification by mass spectrometry.

### 2.4. In-Gel Enzymatic Digestion and Mass Fingerprinting

Protein spots of interest were manually excised from Coomassie stained SDS-PAGE gels and de-stained before digestion. De-stained spots were washed once with ultrapure water, then flushed with 50 mM ammonium bicarbonate (AMBIC: NH_4_HCO_3_)–ethanol (1:1, *v*/*v*). Gel plugs were dehydrated with absolute ethanol and reduced with 10 mM DTT in 50 mM AMBIC for 1 h at 37 °C and alkylated with 55 mM iodoacetamide (IAA) in 50 mM AMBIC for 30 min at room temperature. Before digestion gel pieces were washed with 50 mM AMBIC and dehydrated completely. An aliquot of 0.01 µg/µL trypsin (Promega) was added and proteins were digested overnight at 37 °C. The reaction was stopped by adding 1% (*v*/*v*) trifluoroacetic acid (TFA) in water.

#### Matrix-Assisted Laser Desorption/Ionization Time-of-Flight Mass Spectrometry (MALDI-TOF/MS) Analysis

Peptides from digestion were passed through a C_18_ ZipTip (Millipore, Billerica, MA, USA) and eluted with solution of 0.5 mg/mL α-cyano-4-hydroxycinnamic acid dissolved in acetonitrile and 0.1% (*v*/*v*) TFA in water on a Ground Steel MALDI Target plate (Bruker-Daltonics) precoated with a layer of 10 mg/mL α-cyano-4-hydroxycinnamic acid dissolved in ethanol–acetonitrile–0.1% (*v*/*v*) TFA in water (49.5:49.5:1). Utraflex III MALDI TOF/TOF mass spectrometer (Bruker-Daltonics, Bremen, Germany), operated in reflector mode was used to acquire mass spectra. The Peptide Calibration Standard (Bruker-Daltonics) was used for external calibration. Obtained spectral data were processed by FlexAnalysis software v. 3.4 (Bruker-Daltonics). Internal calibration was acquired from porcine trypsin autolysis peaks (Trypsin Gold, Promega). After exclusion of contaminant ions, a database search was performed using the MASCOT 2.4 algorithm (www.matrixscience.com) against the NCBIprot (National Center for Biotechnology Information) database restricted to “mammalia” taxonomy with following parameters: up to two missed cleavage sites were allowed for trypsin, 50 ppm as maximal peptide tolerance, carbamidomethylation of cysteines as fixed modification and oxidized methionines as the variable. Proteins with scores greater than 76 were considered significant (*P* < 0.05) and confidently identified. The tandem mass spectrometry (MS/MS) analysis was performed in LIFT mode and each precursor ion was selected manually. Spectra pre-processing were performed by FlexAnalysis v. 3.4 (Bruker-Daltonics). Protein database search was performed using the following criteria: up to two trypsin missed cleavages were allowed, precursor ions mass tolerance was set to 75 ppm and fragment ions to 0.6 Da, respectively; cysteines carbamidomethylation and methionine oxidation were selected as fixed and variable modification. The taxonomy was restricted to “mammalia.” Individual ion scores >48 were considered as significant matches (*P* < 0.05).

## 3. Results and Discussion

### 3.1. Sperm Parameters in Fresh and Liquid Stored Semen

Evaluation of semen quality is an important step while selecting the samples to be used for AI. This selection depends on the evaluation of basic sperm parameters although the clinical value of these analyses is still debated [16]. In the present study, 10 boar ejaculates were analyzed on the day of collection and after three days of storage. Table 1 shows semen parameters of fresh and liquid-stored samples. The average sperm concentration was 295.9 ± 74.4 × 10^6^/mL. 

During the first 72 h of in vitro storage the quality of liquid preserved boar spermatozoa is reduced. Spermatozoa respond to preservation-induced stress differently and this can be observed as cell death or sub-lethal damage. Cell death can be assessed by microscopy or flow cytometry techniques and is usually associated with a loss of motility and/or membrane integrity. With the use of computer-assisted semen analysis (CASA), a variety of standardised kinetic parameters can be determined [3,5]. To evaluate the fertilizing ability of semen it is vital to assess concentration, motility and morphology of sperm cells. Among the characteristics mentioned, the motility characteristics of spermatozoa are one of the most important semen parameters associated with the fertilizing capacity [17,18]. Table 1 shows that motility, progressive motility, normal morphology and viability significantly decreased after three days of liquid storage (*P* < 0.05). However, all the investigated ejaculates would normally be considered as suitable for AI since a minimum of 60–70% motile sperms is needed for AI in pigs to result in normal fertility outcome [19]. It was previously shown that reproductive technologies such as washing spermatozoa through a density gradient and pelleting the sample induce in vitro capacitation; on the other hand, freeze-thawing procedures cause capacitation-like changes on the spermatozoon surface [20]. This is probably the main reason why in our study the percentage of non-capacitated spermatozoa also significantly decreased during storage (*P* < 0.05). This deterioration of sperm function has a clear implication for AI practice. Indeed, to fertilize the egg successfully, a given spermatozoon needs to capacitate in a close time relationship to ovulation [19]. Furthermore, we observed a significant increase of acrosomal reacted spermatozoa in liquid stored sperm (*P* < 0.05). In boars, changes in the acrosome are often a result of damaged membranes and degenerative acrosome exocytosis. 

### 3.2. Two-Dimensional (2-D) Electrophoresis and Mass Analysis

Although conventional semen analysis provides valuable information, it was the focus of the present study to investigate potential quantitative differences at the protein level. In this view, a proteomic approach was used to evaluate changes between fresh and liquid stored spermatozoa protein profiles. Two-dimensional gel electrophoresis profiles of spermatozoon proteins extracted from fresh and liquid stored semen showed a high number of protein spots detected along the pH and molecular weight range of the gel (Figure 1). Only matched spots detected on all images in two compared groups (fresh and liquid-stored sperm) were considered for image analysis.

Comparative quantitative image analysis of 2-D gels by Progenesis SameSpot revealed four proteins that significantly differed in their abundance within the analysed groups. Interestingly, three out of four proteins were more abundant in liquid stored samples when compared to fresh sperm protein profile (Table 2). 

Three out of four protein spots (241, 568, 644) were also successfully identified through mass spectrometry analysis and the MS/MS identification confirmed these proteins as belonging to *Sus scrofa.* Protein identities are presented in Table 3. 

The excised protein in spot 241 was identified as ATP citrate lyase (ACL) with the Mascot score of 131 and was over-expressed in the liquid-stored sperm (Table 2 and Table 3). MS analysis showed the theoretical mass of this enzyme to be 120.7 kDa and its isoelectric point 7.4. We obtained 20 matched peptides that covered 23.8% of the protein sequence (Table 3). ACL is a primary enzyme responsible for the synthesis of cytosolic acetyl-coenzyme A (acetyl-CoA) and has a central role in de novo lipid synthesis. It is responsible for catalysing the conversion of citrate and CoA into oxaloacetate and acetyl-CoA along with the hydrolysis of ATP [21]. Acetyl-CoA is later used for fatty acid and cholesterol synthesis [22]. Novak and co-workers [23] reported a positive relationship of horse spermatozoa citrate lyase with fertility. They proposed that increased sperm metabolism and the ability to use carbohydrates as energy may have a positive effect on the fertilizing ability of sperm. It was reported by Chaturvedi and Rama Sastry that the choline acetyl-transferase-acetylcholine-acetyl-cholinesterase system plays a significant role in effective sperm motility [24]. Mammalian semen contains citric acid which may serve as a source for the synthesis of acetyl-CoA by ATP-citrate lyase in spermatozoa. In the present study, increased expression of this enzyme in the extracts of liquid stored spermatozoa was observed. Our results could be explained with spermatozoa capacitation-like changes since the values of capacitated-like spermatozoa were significantly increased in stored semen although spermatozoa exhibited diminished motility and viability (Table 1). A study published by Chen and co-authors proposes that the increase in glycolytic enzymes in frozen-thawed boar sperm is a cell stress reaction to low temperature, or a response to increased demand for energy in the hyper-activated state of premature capacitation in spermatozoa [25]. We can propose that the over-expression of ATP citrate lyase that was observed in stored spermatozoa during our research may have a physiological role in capacitation-like changes, acrosome reaction, and sperm metabolism related to liquid preservation.

The protein in spot 568 was identified as chaperonin containing TCP1 subunit epsilon with the Mascot score of 49 and was found over-expressed in liquid stored sperm samples (Table 2). Matrix-assisted laser desorption/ionization time-of-flight mass spectrometry (MALDI TOF/MS) analysis showed the theoretical mass of this protein to be 60.1 kDa and its isoelectric point 5.7. We obtained seven matched peptides that covered 18.0% of protein sequence (Table 3). Chaperonins use energy derived from ATP hydrolysis to assist protein folding. Chaperonin containing the T-complex/TCP1-ring complex (CCT/TRiC) is a hetero-oligomeric complex enclosing a central cavity that is able to bind denatured or unfolded polypeptides. Each ring is composed of eight (CCT1–CCT8) unique subunits. Subunits have molecular mass between 52 and 65 kDa and are essential for the chaperoning activity. The T-complex is known to harbour genes that affect mouse development and male fertility. Within the testes, this complex is believed to be crucial for the morphological differentiation of spermatids. Dun et al. [26] demonstrated that an intact CCT/TRiC complex is predominantly expressed within the peri-acrosomal region of the sperm head of mature human and mouse spermatozoa. The complex is present in testicular mouse sperm but during capacitation undergoes significant changes in subcellular localization, including an increase in the surface expression of at least two CCT subunits. Furthermore, it has been shown that CCT/TRiC participates indirectly in sperm–zona pellucida (ZP) adhesion of human and mice spermatozoa because of its ability to form a stable complex with ZP receptors, including ZPBP2 (zona pellucida-binding protein 2) [26]. Proteins of the CCT/TRiC complex were also found as a part of bull sperm surface proteome [27]. D’Amours and co-workers [28] reported that the subunits 5 and 8 of the T-complex protein 1 (CCT) were found in a higher concentration in the low-fertility bull sperm samples. In frozen-thawed boar semen, an increase of proteins related to sperm-oocyte binding and fusion, among them chaperonin containing T-complex polypeptide 1 (TCP1), subunit 7 (eta) was observed by Chen and co-workers [25]. We also found that the T-complex protein 1 subunit epsilon was over-expressed in liquid-stored semen which exhibited lower motility and viability characteristics and a greater percentage of capacitated-like and acrosome-reacted spermatozoa compared to fresh semen. 

The protein in spot 629 was also over-expressed in the liquid-stored boar sperm but was not identified with an appropriate certainty (Table 2 and Table 3). The MS database search using the MASCOT 2.2.03 algorithm resolved the protein to be a probable phospholipid-transporting ATP-ase, but from a different animal species (horse). The MS/MS ions search did not confirm the obtained result. Phospholipid-transporting ATP-ases transport phospholipids across the membrane. Significant plasma membrane restructuring, e.g., phospholipid scrambling (a reduction in phospholipid asymmetry through movement of one or all four phospholipids both inward and outward across the membrane lipid bilayer) appears to occur upon capacitation. In vitro capacitation also induces sperm surface changes in the composition and topology of sperm plasma membrane lipids and proteins [20]. Wang and co-workers [29] discovered an ATP-dependent aminophospholipid transporter, exclusively expressed in the acrosomal region of mouse spermatozoa named ”Phospholipid-transporting ATP-ase IK” that is critical for normal phospholipid distribution in the bilayer, and for normal binding, penetration, and signaling by the zona pellucida. A chromosome X encoded phospholipid–transporting ATP-ase was previously also found as a part of a bull sperm surface proteome [27]. The identity and the reason for higher expression of the probable phospholipid-transporting ATP-ase in boar spermatozoa after liquid preservation needs to be elucidated, and thus further studies targeting the role of this protein are desirable. 

The protein in spot 644 was identified with the Mascot score of 83.8 as cytosolic non-specific dipeptidase that belongs to the peptidase M20A family and was the only one of the found proteins that was over-expressed in fresh boar sperm (Table 2). Its theoretical mass was determined to be 53.1 and its isoelectric point 5.2. We obtained six matched peptides that covered 9.3% of protein sequence (Table 3). Cytosolic non-specific dipeptidase is found in the cytoplasm and hydrolyses a variety of dipeptides, preferentially hydrophobic dipeptides. Its presence in human seminal fluid has been reported previously [30]. Cytosolic non-specific dipeptidase was among proteins that were found in significantly different amounts in human spermatozoa samples before and after freezing and thawing [31]. Recently, in-depth proteomic analysis of boar spermatozoa through shotgun and gel-based approaches identified more than 2000 proteins among which cytosolic non-specific dipeptidase was also found [32]. The function of this enzyme and the reason for its decreased expression in liquid-stored boar spermatozoa is not yet determined and should be further studied. 

## 4. Conclusions

The proteomic approach employed in this study revealed differences in expression of four proteins between fresh and liquid-stored boar spermatozoa. ATP citrate lyase, chaperonin containing TCP1 subunit ε and probable phospholipid-transporting ATP-ase were over-expressed in stored sperm while cytosolic non-specific dipeptidase was over-expressed in fresh sperm samples. Since most of them have a known or at least probable physiological role in capacitation, acrosome reaction and sperm metabolism, the identified proteins could be used as indicators of boar sperm quality and spermatozoa survival after liquid storage. The generated dataset will be useful in developing appropriate strategies for efficient semen preservation in the swine industry.

## Figures and Tables

**Figure 1 animals-10-00553-f001:**
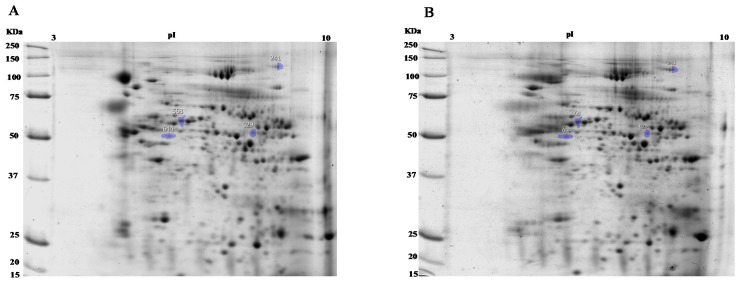
Representative 2-D electrophoresis map of boar spermatozoa extracts. (**A**) Fresh boar spermatozoa extracts, (**B**) liquid stored boar spermatozoa extracts. Differentially expressed proteins are indicated in the map. KDa: protein molecular weight expressed in kilodalton; pI: isoelectric point.

**Table 1 animals-10-00553-t001:** Sperm characteristics (mean ± standard deviation (SD)) of fresh semen and semen after 72 h of liquid preservation at 15–17 °C.

Semen Trait	Fresh Semen	Stored Semen	*P*-Value
motility (%)	85.7 ± 4.3	70.7 ± 6.7	*P* < 0.05
progressive motility (%)	49.6 ± 8.7	35.4 ± 3.7	*P* < 0.05
normal morphology (%)	76.4 ± 8.7	64.6 ± 10.3	*P* < 0.05
viability (%)	92.4 ± 4.6	70.2 ± 10.2	*P* < 0.05
non-capacitated spermatozoa (%)	83.9 ± 6.0	60.2 ± 8.6	*P* < 0.05
acrosomal reacted spermatozoa (%)	3.6 ± 2.0	14.3 ± 6.0	*P* < 0.05

**Table 2 animals-10-00553-t002:** Differences in expression of 4 spots between fresh and liquid stored sperm.

Spot No.	Paired *t*-Test (*P*)	Fold	Average Normalised Volumes	
			Fresh Semen	Stored Semen
568	0.019	1.3	1.150 × 10^7^	8.745 × 10^6^
644	0.032	1.3	7.770 × 10^6^	5.920 × 10^6^
629	0.033	1.3	3.768 × 10^6^	4.937 × 10^6^
241	0.046	1.2	3.562 × 10^6^	4.429 × 10^6^

**Table 3 animals-10-00553-t003:** Differentially expressed proteins identified by 2-DE/MS and confirmed by tandem mass spectrometry (MS/MS).

Spot No.	Entry Number	Description	MS				MS/MS			
			Theoretical M_r_(kDa)/pI	Mascot Score (†)	% Seq. Coverage	Matched Peptides	Peptide Sequence	Mascot Score (‡)	m/z	z
241	gi/341942463	ATP citrate lyase	120.7/7.4	131	23.8	20	WGDIEFPPPFGR	52	1417.690	1
568	gi/523580068	chaperonin containing TCP1 Subunit 5 (epsilon)	60.1/5.7	49	18	7	WVGGPEIELIAIATGGR	96	1738.949	1
629		Not identified								
644	gi/343962597	cytosolic non specific dipeptidase	53.1/5.2	83.8	9.3	6	ALQTVFGVEPDLTR	75	1545.827	1

^†^ Protein scores greater than 76 are significant (*P* < 0.05). ^‡^ Individual ion scores > 48 indicate identity or extensive homology (*P* < 0.05). Database search in NCBI: National Center for Biotechnology Information; Taxonomy: Mammalia; Precursor ion error: 75 ppm; Fragment ion error: 0.6 Da. ATP: adenosine triphosphate, TCP1: T-complex polypeptide 1.

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
