# Peer review of "Proteomic Analysis of Fresh and Liquid-Stored Boar Spermatozoa"

_animals, 2020, doi:10.3390/ani10040553_

Round 1

Reviewer 1 Report

This work aims to explore proteomic profile of fresh and liquid stored for three days boar sperm. The storage time of three days is widely used for traditional AI. However, longer storage time (5-7days and more than 10 days) may cause a much difference and I don’t know why authors just treat boar semen only for 3 days. Second, for different storage time, different extenders are also needed which results in different results.

-Some sentences are very short which can be joined with adjacent sentences to improve the write up for example “L98- Semen was then stored for three days at 15–17 °C.”

- How and why did you prepare the semen aliquots from 4 pig breeds? In what combination among the individuals?

-10 semen samples in total or from one animal? Sampling and pool preparation from samples are not clear!

- “Gel electrophoresis” the whole paragraph needs to be re-written as it is difficult to understand.

-Figure 1. the figure should be self-explanatory; what do 3, pI and 10 stands for? The whole figure is not clear and readable.

Minor Comments

-L34-L35- rewrite the sentence to make it more understandable

“Semen from eight boars was analyzed both fresh and after liquid preservation at 15–17 oC for three days”

-L39-L43- This sentence is difficult to decipher; it would be better if it is divided into 2 separate sentences.

-L59- seminal volume, sperm concentration…

-L61- litter size [7], but may not accurately…

-L64- “among individuals or from individual to individual”. “Significant individual variations were observed in”

-L79- may also help to discover…

Materials and Methods

-L95- following sentence should be revised for better presentation “To assess sperm viability Hoechst staining was used [11] and Giemsa-staining to assess semen morphology [12].”

-L105-L106- both sentences should be revised

-L108- “pellets were washed…” it’s better to write plural sentence

-L114- “and” phosphatase inhibitor cocktail

-L117- 14000 or 1400 × g?

-L122- read should be replaced with measured

-L133- “30 min at 2500 V and at 4000 V until 40.000 V/hours were reached in total” protocol should be revised to make it clearer.

-L195- should be “observed as cell death” instead of observed as a cell death

-L204- 60–70% motile sperms are needed…

-L210-L212- sentence should be revised in accordance with grammar and arrangement.

-L217-L218- changes between fresh and stored spermatozoa protein profiles…

-L292- because of its ability…

-L296-L298- this sentence should be revised

Author Response

Thank you for your comments and suggestions. Please see the attachment with provided response.

Reviewer 2 Report

In this study, Premrov Bajuk et al. compare the proteomic profile between fresh and stored boar semen. They found that 4 protein spots intensities differ between fresh and stored semen. While the topic is interesting, there are some concerns mainly about the methodology and the statistical analyses employed.

– In general, the sperm analyses need to be described more in depth:

  • Lines 92-95. The CASA settings need to be provided. How many fields were examined per sample? What was the cut-off to define a motile sperm?
  • Lines 96-97. What type of microscope, objective, and magnification were used? How many sperm were analyzed per sample?
  • Line 96. Sperm morphology instead of semen morphology.

– The authors must include a statistical analyses section at the end of the materials and methods and add all the statistical analyses of the study. The statistical significance (p value) of the sperm parameters between groups has to be included in Table 1.

  • Lines 100-102: The authors must check the normal distribution (Shapiro–Wilk test) and homogeneity of variance (Levene’s test) of all the sperm variables. Afterwards, a repeated measures test needs to be performed (paired t-test or Wilcoxon signed-rank test, depending on the distribution of the data). Authors also need to include if the data are expressed as mean or median and SD or SEM.
  • Line 146. The use of One-way Anova test is not appropriate here because the authors compare two groups (fresh and stored semen). A repeated measures test needs to be used instead (paired t-test or Wilcoxon signed-rank test).

– Lines 6-16: Two authors’ affiliations are repeated twice (1 and 3).

– Lines 120-123. More information is needed in this section (method used, standard curve concentrations, etc.).

– Lines 182. The authors did not make any correlation analyses. The title of this section has to be changed.

The following article PMID: 31444576 might be useful for the discussion of the results obtained and to introduce the topic.

Author Response

(The authors gave the same response as above.)

Reviewer 3 Report

In this manuscript entitled “Proteomic Analysis of Fresh and Liquid Stored Boar Spermatozoa” submitted by Premrov Bajuk et als, described proteomic changes along the conservation period in comparison with fresh sample. MM and Results sections are done correctly, however some changes must be performed before to be considered for publication. Discussion section is too long and moreover some concepts are misunderstood by the authors. My principal concern is that they confused the terms capacitation and capacitation-like changes associated to preservation. I encourage the authors to read DOI:10.1002/mrd.22663; DOI: 10.1111/j.1439-0531.2011.01799.x and DOI:10.1016/j.theriogenology.2016.09.046 and rewrite the discussion section. Given the authors evident knowledge of proteomic field, I feel they could make a lot more of this section and produce a carefully research manuscript that would be valued by their peers. 

Other minor comments:

Authors should focus more in the abstract and introduction section on bibliography/results about preservation at 17ºC and not talk too much about cryopreservation since them didn´t do nothing related to cryopreservation.

Why the authors choose to perform the experiments at day 3 of conservation?

Line 173-174- Please could you clarify why did you set Proteins with scores greater than 76 as significant and confidently identified.

CASA section should be improved and detailed, for instance how much time the sperm were warmed before analysis? Were they warmed? Which one were the settings for CASA measurement?

Capacitation is a serial of event that happens during their transit along the female tract and these conditions can be mimic in vitro using BSA and bicarbonate activating sAC/PKA pathway. Since, you are not stimulating sperm capacitation please reconsider use capacitation-like changes instead of capacitation.

Table 1- Please include the statistic in the table.

Do the author have access to reproductive outcome of AI performed at day 0-1 and day 3 of conservation. Does the reproductive outcome differ between days of conservation?

Line-194 please eliminate “in the ejaculate”

Please eliminate line 207 to 208 “Cryopreservation changes the functional state of many proteins including the 208 enzymes related to sperm metabolism, proteins related to capacitation and acrosome reaction [19]”. Authors should focus on comparison with preserved not with cryopreserved spermatozoa.

Please insert in Figure 1, two different 2D-gel comparing where we can see the spot in fresh and in preserved conditions.

Line265-276 should be eliminated. Authors don´t capacitate spermatozoa authors should consider re-write the discussion having in mind capacitation-like changes induced by preservation. Please see DOI: 10.1111/j.1439-0531.2011.01799.x and DOI: 10.1071/rd01113

Discussion section should be concise and discern appropriately the terms: “capacitation” and “capacitation-like changes induced by preservation”

Author Response

(The authors gave the same response as above.)

Round 2

Reviewer 1 Report

The authors revised carefully and improve a lot in present form. I am satisfied with this revision.

Reviewer 2 Report

The authors addressed all comments.

Reviewer 3 Report

Authors follow reviewer suggestions. They amended the text and improved the figure. I don´t have further suggestion.